

# Rat liver regeneration following ablation with irreversible electroporation

Alexander Golberg[1,2], Bote G. Bruinsma[1,3], Maria Jaramillo[1], Martin L. Yarmush[1,4] and Basak E. Uygun[1]

[1] Center for Engineering in Medicine, Massachusetts General Hospital, Harvard Medical School, and Shriners Hospitals for Children in Boston, Boston, MA, United States
[2] Porter School of Environmental Studies, Tel Aviv University, Tel Aviv, Israel
[3] Department of Surgery (Surgical Laboratory), Academic Medical Center, University of Amsterdam, Amsterdam, The Netherlands
[4] Department of Biomedical Engineering, Rutgers University, Piscataway, New Jersey, United States

## ABSTRACT

During the past decade, irreversible electroporation (IRE) ablation has emerged as a promising tool for the treatment of multiple diseases including hepatic cancer. However, the mechanisms behind the tissue regeneration following IRE ablation have not been investigated. Our results indicate that IRE treatment immediately kills the cells at the treatment site preserving the extracellular architecture, in effect causing in vivo decellularization. Over the course of 4 weeks, progenitor cell differentiation, through YAP and notch pathways, together with hepatocyte expansion led to almost complete regeneration of the ablated liver leading to the formation of hepatocyte like cells at the ablated zone. We did not observe significant scarring or tumor formation at the regenerated areas 6 months post IRE. Our study suggests a new model to study the regeneration of liver when the naïve extracellular matrix is decellularized in vivo with completely preserved extracellular architecture.

## INTRODUCTION

Irreversible electroporation (IRE) was proposed for non-thermal tissue ablation a decade ago (*Davalos, Mir & Rubinsky, 2005*). It is now an emerging tool in the interventional radiology for solid tumors ablation and has other potential medical uses in hemostasis and the treatment of wounds (*Golberg & Yarmush, 2013*; *Golberg et al., 2013*; *Golberg et al., 2014*; *Golberg et al., 2015a*; *Yarmush et al., 2014*). Electroporation leads to an increase in cell membrane permeability following exposure to high-voltage, pulsed electric fields (*Weaver & Chizmadzhev, 1996*). In IRE, the change of permeability is irreversible and results in cell death (*Miller, Leor & Rubinsky, 2005*; *Golberg & Yarmush, 2013*).
The distinguishing characteristic of IRE from other tissue ablation methods is that it only affects the cell membrane, preserving the structure of the extracellular matrix (ECM) which results in more rapid restoration of tissue perfusion (*Golberg et al., 2013*). However, the cellular mechanisms of IRE-induced death remains unclear: in vivo ablation is

Corresponding authors
Alexander Golberg, agolberg@gmail.com
Basak E. Uygun, buygun@mgh.harvard.edu

currently thought to result from a combination of mechanisms including necrosis, caspase-dependent (apoptotic) and caspase-independent cell death (*Song et al., 2014*; *Beebe, 2015*). What is even less known, is the response of tissues to the pulsed electric fields treatment. In prior reports, we showed IRE-ablated rat skin regenerates without scars (*Golberg et al., 2013*; *Golberg et al., 2015a*). An interesting clinical observation after IRE treatment of liver tumors is that the liver tissue assumes its normal structure much more quickly than following thermal ablation, as assessed by imaging methods. It is speculated that the reduced inflammation and fibrosis after the IRE treatment may be the underlying cause (*Narayanan, 2011*). Intrigued with this clinical observation, we set out to investigate liver histologically following IRE ablation to determine the cellular events constituting the tissue response.

The liver has tremendous restorative capacity and is able to regenerate after removal of 90% of organ mass (*Michalopoulos, 2010*) and different regenerative mechanisms have been identified in various injurious conditions. Liver regeneration after hepatectomy involves expansion of remaining hepatocytes and other liver cells to restore the initial organ mass. In the case of chronic liver injury, resident hepatic progenitor cells become active and start differentiating to replace the injured hepatocytes. The resident progenitors, sometimes called "oval cells" or "small hepatocytes," emerge from the bile ducts or canals of Hering (*Oertel & Shafritz, 2008*; *Turner et al., 2011*; *Best et al., 2015*) and may differentiate into both hepatocytes and biliary cells (*Español-Suñer et al., 2012*; *Huch et al., 2013*). However, the complex signaling pathways (*Taub, 2004*; *Fausto, Campbell & Riehle, 2006*; *Michalopoulos, 2007*; *Böhm et al., 2010*) that regulate the fate of differentiating cells in the regenerating liver are not completely clear and the relations between "oval cells," bile ducts cells and hepatocytes are debated in the literature (*Greenbaum, 2011*; *Michalopoulos, 2012*).

Here we show that liver regeneration following IRE ablation occurs through progenitor cell activation as well as hepatocyte expansion. Different from the currently used models for liver regeneration studies such as partial hepatectomy and chemical-induced liver injury, IRE kills the cells by damaging the cell membrane, preserving the ECM and tissue perfusion. We show that the preserved ECM serves as a substrate for the development of mature liver tissue. In addition, our findings suggest the use of IRE ablation as a new model to study liver regeneration.

## MATERIALS AND METHODS

### Animal subjects

Female Lewis rats (N = 21, 180–250 g, 6-weeks old) were obtained from Charles River Laboratories (Wilmington, MA, USA). The animals were housed in individual cages with access to food and water ad libitum, and were maintained on a 12-hour light/dark cycle in a temperature-controlled room. All animal procedures were described in protocol number 2013N000127 and were approved by the Institutional Animal Care and Use Committee (IACUC) of the Massachusetts General Hospital and were kept in accordance with the guidelines of the National Research Council.

## Rat liver irreversible electroporation

Twenty minutes prior to induction of anesthesia, 0.05 mg/kg of buprenorphine was administered subcutaneously. Rats were anesthetized by inhalation of isoflurane (5% in 100% oxygen) in an Ohmeda Tech 4 tabletop anesthesia apparatus connected to a standard rodent system. The animal was placed in a supine position on a sterile surgical table. The abdomen was opened with a 25 mm midline incision from the xyphoid process down and the sternum and abdominal wall were retracted. Two non-thermal IRE ablations were performed on different lobes using small, 7 mm (diameter) BTX Tweezertrodes (Harvard Apparatus, Holliston, MA, USA), which reduced the concentration of the current on electrodes tips, rapid drops in the electric fields and edge thermal effects associated with the needle electrodes. In the previous work, using plate electrodes with 1 cm$^2$ surface area, we ablated rat liver by applying 90 unipolar, rectangular electric 70 $\mu$s pulses, at 150 Vmm$^{-1}$ potential difference between the electrodes, at 4 Hz. That protocol led to 0.5 °C increase of liver temperatures after the IRE procedure (*Golberg et al., 2011*). The protocol used in this study was optimized in our most recent work that investigated the electric field distribution in the rat liver (*Golberg et al., 2015b*).The electroporation parameters used in this study were as follows: electrode separation: 3 mm, applied voltages: 360 V, number of pulses: 99, pulse length 50 $\mu$s, frequency of pulse delivery: 4 Hz. Square pulses were delivered using BTX 830 pulse generator (Harvard Apparatus Inc, Holliston MA, USA). Currents were measured in vivo using PicoScope 4224 Oscilloscope with Pico Current Clamp (60A AC/DC) and analyzed with Pico Scope 6 software (Pico Technology, Cambridgeshire, UK). The abdomen was irrigated with 0.9% saline solution and the incision was closed in two layers using 4-0 silk sutures.

## Histology

Tissue specimens were harvested immediately after IRE and 24 h, 3 d, 7 d, 14 d, 28 d and 6 mo (N = 3 for each time point up to 6 months and N = 6 for a 6 months time point) following the initial IRE ablation. Liver samples were fixed in 10% formalin, embedded in paraffin, and cut into 5 $\mu$m sections. Sections were stained with hematoxylin and eosin (H&E) and Masson's trichrome stains by the Pathology Core at the Massachusetts General Hospital. Slides were evaluated by three separate investigators. Color images of each entire tissue section were acquired using NanoZoomer Digital Pathology System (Nanozoomer 2.0-HT slide scanner; Hamamatsu, Hamamatsu City, Japan).

## Immunohistochemistry

Paraffin-embedded tissue sections (5 $\mu$m) on glass slides were baked at 60 °C for 30 minutes, followed by deparaffinization in xylene and rehydration in graded alcohol into water. Antigen retrieval was performed by boiling the slides in 10 mM sodium citrate buffer pH 6.0 for 30 minutes. Endogenous peroxidase activity was quenched with Dual Endogenous Enzyme Block (DAKO, Glostrup, Denmark) for 5 minutes. Tissue sections were incubated with 1:100 dilution of Ki67 rabbit monoclonal antibody (Abcam, Cambridge, MA, USA), 1:1000 dilution of Sox9, rabbit polyclonal antibody (Millipore, Bedford, MA, USA), 1:2000 dilution of $\alpha$-smooth muscle actin ($\alpha$-SMA) rabbit

monoclonal antibody (Millipore, Bedford, MA, USA) in 1% TBS/BSA at room temperature inside a humidified chamber for 30 minutes. After washing, slides were incubated with rabbit polymer reagent (DAKO, Glostrup, Denmark) for 30 min at RT followed by incubation with the DAB+ reagent (DAKO, Glostrup, Denmark) with monitoring for 5–10 minutes. After washing, counterstain was done using Harris type hematoxylin. Slides were briefly dehydrated and then mounted with Histomount solution (Life Technologies, Grand Island, NY, USA). Color images of each entire tissue section were acquired using NanoZoomer Digital Pathology System (Nanozoomer 2.0-HT slide scanner; Hamamatsu, Hamamatsu City, Japan). The number of cells that are positive for Ki67 was counted using the software ImageJ (NIH, MD, USA).

### q-RT-PCR

Biopsies (25–30 mg) of liver tissue were used to extract RNA using the NucleoSpin kit (Macherey-Nagel, Bethlehem, PA, USA) according to manufacturer protocols. Absorbance at 280 nm and 260 nm was measured using a NanoDrop 100 to obtain RNA concentration and quality. Reverse transcription was performed using an ImProm II reverse transcription kit (Promega, Madison, WI, USA) following manufacturer recommendations. qRT-PCR analysis was performed in ViiA7 instrument (Applied Biosystems, Grand Island, NY, USA) using SybrGreen system and the primers listed in Table 1.

## RESULTS

### Pulsed electric fields cause immediate vascular congestion and rapid decellularization in the exposed area

Pulsed electric fields were applied on separate lobes of the liver through the two flat circular electrodes (Fig. 1A). Immediately after the ablation, the treated sites on the liver lobes lost the normal color and were distinctly white in appearance, as expected (*Jarm et al., 2010*) (Fig. 1C). Histological observation showed the complete congestion of the treated area immediately after the application of electric fields. Trapped red blood cells were observed in the whole treated area (Fig. 1E). The extracellular structure of liver appeared preserved after the IRE. Twenty-four hours after the ablation, the treatment site was completely ablated (Fig. 1F). The large vessels and ductular structures were preserved. The ablated area was congested with cell debris. Rare epithelial and ductular cells survived at the area of the large blood vessels (Fig. 1F, arrows). This survival of cells near the large blood vessels supported the previously developed "electric field sinks" mechanism, which showed that large vessels cause a rapid drop in the electric field strength in areas parallel to the major vector of the electric field because of the difference in electrical conductivities of blood, vessel wall and parenchymal tissues. (*Golberg et al., 2015b*).

### Early events during liver regeneration after IRE ablation

The cells in the treatment site die post IRE treatment. Regenerating areas appear within the site as early as 3 d (Fig. 2A). These areas are composed of non-homogenous cell populations with elongated and migratory cells (black arrowhead) and ductular

**Table 1 Primers used for q-RT-PCR.**

| Primer | Sequence |
|---|---|
| Ros-1 F | agaaagacgagaggcaacca |
| Ros-1 R | aggagagccataagccacct |
| Cdh22 F | catcgctctcttggtctgtg |
| Cdh22 R | gtcataagcctcggtgtcct |
| Cldn7 F | acagcccctccacttcttg |
| Cldn7 R | aaccgagccaaacacaaatc |
| Muc1 L | caatggcagtagcggtctct |
| Muc1 R | gtgggtagggtgacttgctc |
| Cd133 F | tttgtatgtgccgttgctgt |
| Cd133 R | cgagtccttgtctgctggtt |
| Sox9 F | agagaacgcacatcaagacg |
| Sox9 R | tctggtggtcggtgtagtca |
| Tgfα F | cgcagacccaaaccataagt |
| Tgfα FR | cagggaacaacatcaccatct |
| Fgf1 F | cgctggataggagatgaggt |
| Fgf1 R | atggtgcgttcaagacaggt |
| Dlk/Pref1 F | ggaggctggtgatgaggata |
| Dlk/Pref1 R | ggagggaggggttcttagc |
| Ignγ F | tcttcagcaacagtaaagcaaaa |
| Ignγ R | cccagaatcagcaccgact |
| Scf F | cctgttcttgctacccgtga |
| Scf R | gtgttctcgtccccatcatc |
| Sdf1 F | ctttgagggagggtttggag |
| Sdf1 R | tgcggaggaatgacttctgt |
| Fn14 F | cagactcttccaaccacaagg |
| Fn14 R | acctagcttgaggctctctgtct |
| Ctgf F | gctggagaagcagagtcgtc |
| Ctgf R | acaccccacagaacttagcc |
| Notch2 F | cggcttcagtggtatggact |
| Notch R | attcttacagggctcgctca |
| Jag1 F | tctactggtgtgtgcggaag |
| Jag1 R | aatccttgatggggactgtg |
| Hes1 F | ggacggtgaacgactacacg |
| Hes1 R | gcctttccttttgtgcagag |
| Gapdh F | ggcattgctctctcaagacaa |
| Gapdh R | atgtaggccatgaggtccac |
| C-kit F | tccctgtgaagaacactacgg |
| C-kit R | ttgctttggttgtcggattt |
| Afp F | tgttcctcattggctacacg |
| AFp R | gttcacagggtttgcctcat |

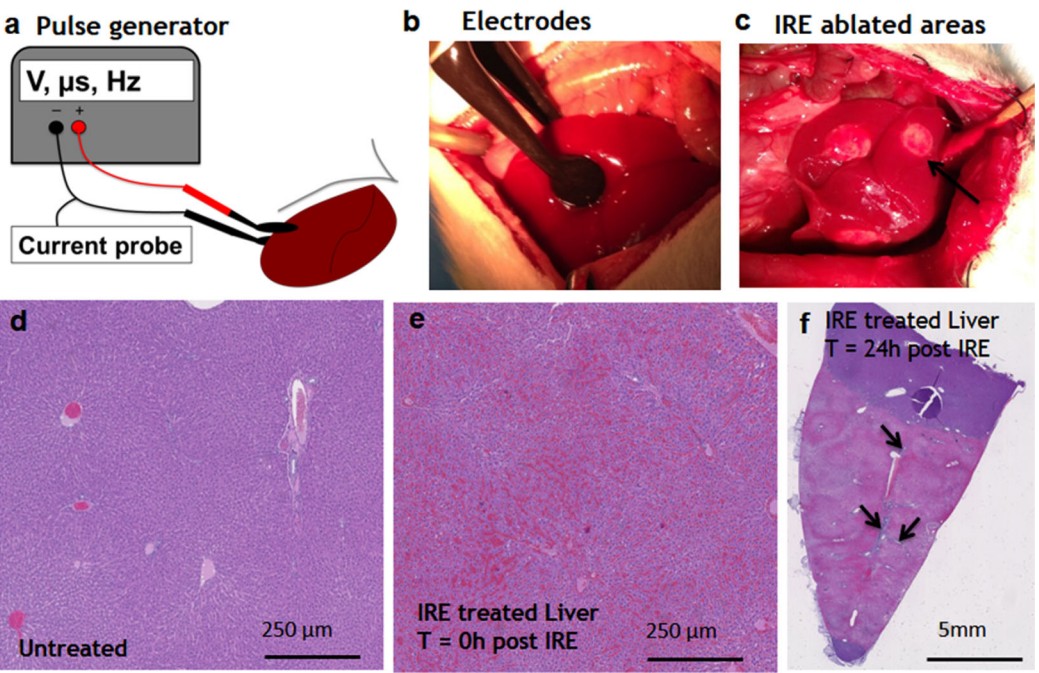

**Figure 1 Ablation procedure with irreversible electroporation.** Irreversible electroporation setup (A) application of the electrodes on the liver lobe (B) ablated areas as appear immediately after the treatment (C). Liver histology (H&E). Normal liver (D) ablated zone immediately after the treatment (E) ablated zone 24 hours after the treatment (F) with areas of rare survived ductal cells (black arrows). N = 3 for each time point.

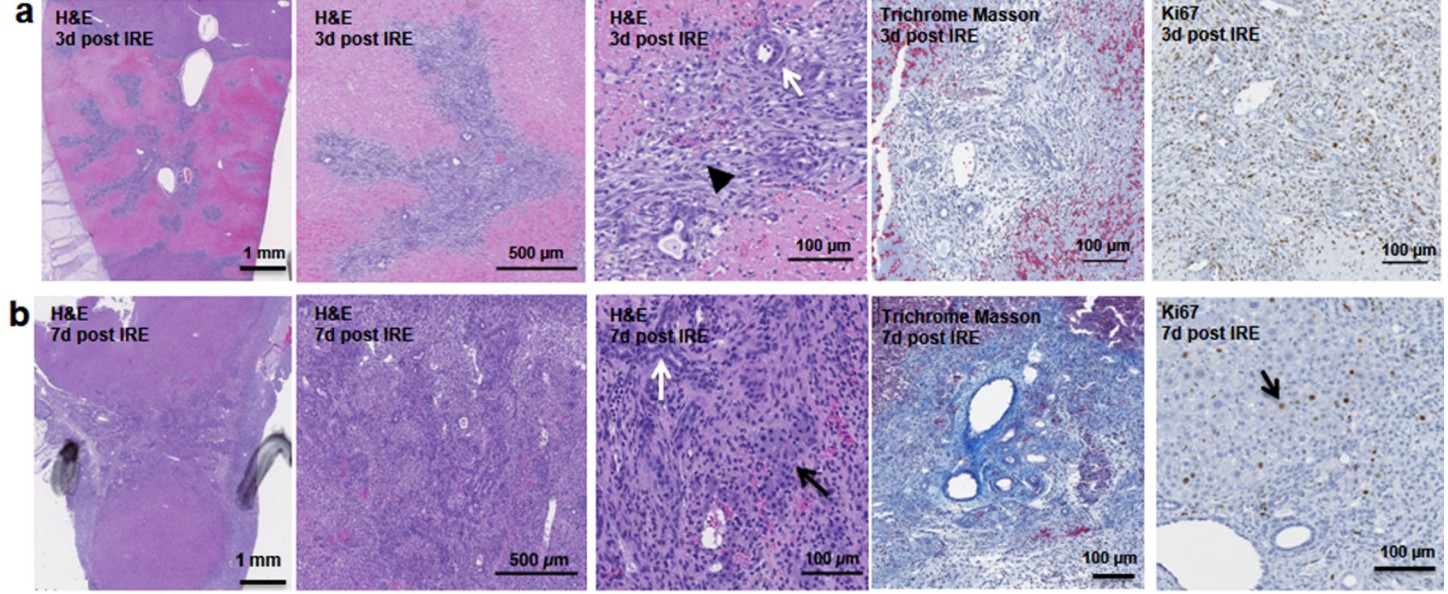

**Figure 2 Early regeneration events following liver irreversible electroporation ablation.** The cells in the treatment site die post-IRE treatment. The regenerative zones appear within the site as early as 3 d (A). The regenerative zones are composed of non-homogenous cell populations, elongated and spindle shapes cells (black arrowhead) and ductular cells (white arrow) (top row). The site is contracted and filled with cells by 7 d (B). The cell population is still heterogeneous in morphology, composed of ductular cells (white arrow), and large cells (black arrow) (bottom row). Extracellular matrix deposition is observed at the treated sites (Masson's trichrome stain). Division of cells, including hepatocytes is observed as indicated with Ki67 stain. N = 3 for each time point.

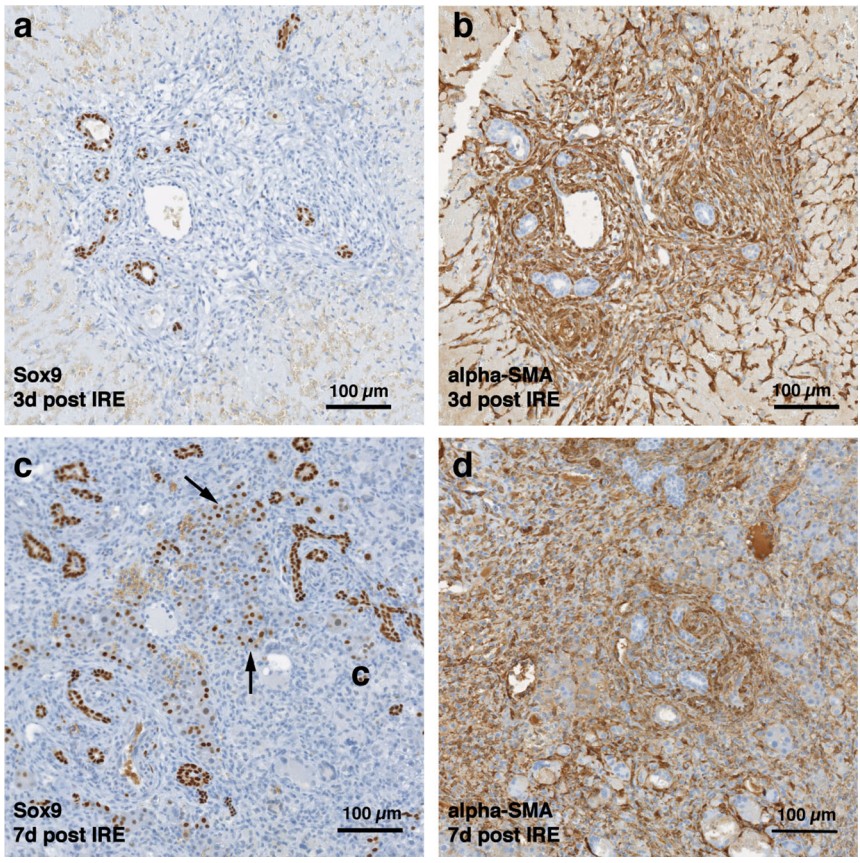

**Figure 3** **Immunohistochemistry of ablated zones-early events.** On day 3 (A, B) and day 7 (C, D) post ablation, Sox9 and alpha-SMA positive cells were observed. 3 days after ablation Sox9 positive cells were all located near the area of the ductular cells (A); however, 7 days after the ablation Sox9 positive cells appear in the parenchymal tissue suggesting migration and differentiation of the ductular cells to the ablated zone (C). Alpha-SMA staining was associated with the non-ductular cells located near the area of vessels and ducts (B, D). N = 3 for each time point.

cells (white arrow) (Fig. 2A). The site area became smaller and filled with cells by 7 d (Fig. 2B). The cell population is still heterogeneous in morphology, composed of ductular cells (white arrow), and large round cells (black arrow). Cells were found actively dividing, as detected with positive Ki67 staining, and they secreted new extracellular matrix (intensive Masson's trichrome staining). Interestingly, 7 d after the IRE ablation some large, hepatocyte-like cells, near the regenerated area were found positive for Ki67, indicating that hepatocyte expansion contributed to the regeneration process.

Starting on day 3, the regenerative spots are filled with elongated cells that are positive for activated stellate cell marker α-smooth muscle actin (α-SMA) (Fig. 3). Ductular cells positive for cholangiocyte marker, Sox9, populated the treatment site suggesting the contribution of progenitor cells to liver regeneration (*Furuyama et al., 2011*; *Kawaguchi, 2013*) (Fig. 3A). By day 7, some larger, hepatocyte-like cells also stained positive for Sox9 (Fig. 3C black arrows). These observations suggest a role for progenitor cells of ductular origins in liver regeneration after IRE ablation.

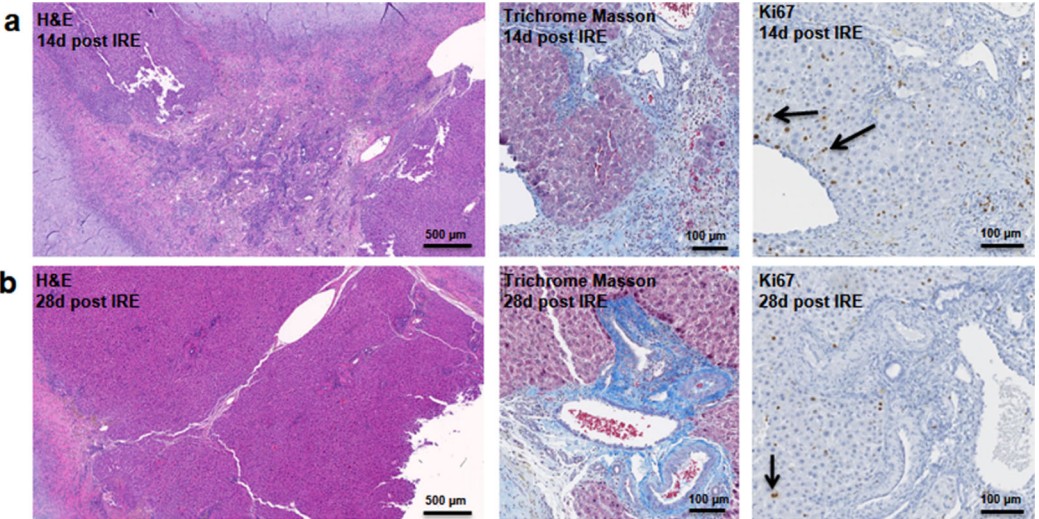

**Figure 4** **Late regeneration events following irreversible electroporation ablation.** 14 days after the ablation, the site is still filled with ductular reactions and multiple proliferating cells (A). The number of proliferating cells reduced at 28 days after the ablation (B) and disappeared 6 months after ablation (Fig. 7). Secretion of extracellular matrix, as detected by Masson's trichrome staining, and active cell division, as detected by Ki67 overexpression, continued at both 14 and 28 days after ablation. Ki67 staining was mostly observed at the dividing hepatocyte-like steps. N = 3 for each time point.

## Progenitor cells and hepatocytes contribute to the liver regeneration after PEF ablation

At two weeks to one month after the ablation, cell proliferation continued near and at the treated site with an appearance similar to ductular reaction. The size of the regenerating area decreased, probably due to contraction induced by stellate cells (*Soon & Yee, 2008*) (Fig. 4). Secretion of the ECM adjacent to the large vessels continued (Fig. 4A). Hepatocyte proliferation at the edge area of the treated zone was shown even one month after ablation (Fig. 4B, Ki67 panel black arrow). Sox9 and α-SMA positive cells were present in all regenerative spots at two weeks and one month after the ablation (Fig. 5). However, the intensity of Sox9 staining in the parenchyma was reduced one month after ablation in comparison with the two weeks' time point (Figs. 5A vs 5B), possibly suggesting the differentiation and maturation of the newly formed hepatocyte-like cells. This suggestion is corroborated by the previous study that showed the downregulation of Sox9 expression associated with hepatogenic differentiation of human liver mesenchymal stem/progenitor cells (*Paganelli et al., 2014*).

To further elucidate some of the possible regeneration mechanisms, quantitative reverse transcriptase polymerase chain reaction was used to analyze the level of expression of markers specific to adult progenitor cells (*Cldn7, Ros1, Cdh22, Cd133, Muc1*) (*Yovchev et al., 2007*) (Fig. 6A), factors implicated in activation of progenitor cells (*Scf, Sdf1, C-kit, Fgf1, Fn14,Dlk1, Tgfα, Ifn*γ) (Fig. 6B), and markers involved in Notch pathway activation (*Notch2, Jag1, Hes1, Sox9*) (Fig. 6C). We observed upregulation of all progenitor specific markers ranging from 5-fold to 2000-fold for *Muc-1* and *Cdh22*, respectively. Similarly, most progenitor cell activation factors except for alpha-fetoprotein (*Afp*)

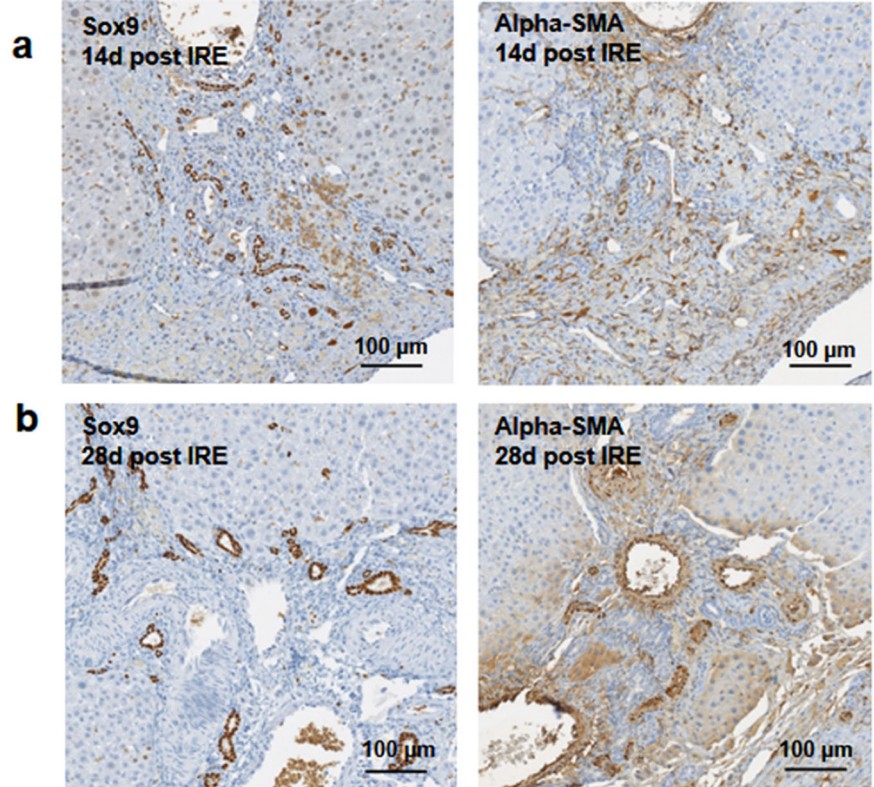

**Figure 5 Immunohistochemistry of ablated zones-late events.** After 14 days of ablation (A, right) multiple cells in the parenchymal tissue expressed Sox9 positive staining. However, 28 days (B) after ablation, the Sox9 positive staining was observed only in the ductular cells as at the 3 days after ablation. Alpha-SMA staining was associated mostly with epithelial cells lining the vessels at both 14 and 28 days after ablation. N = 3 for each time point.

(consistent with immunohistochemistry results) and *Sdf-1* were found to be upregulated with expression ranging from 1.5 to 250-fold with respect to control samples. Finally, all genes related to YAP/Notch pathway activity were also found to be upregulated with average expression ranging from 1.5 to 75 for *Hes1* and connective tissue growth factor (*Ctgf*), respectively. It is important to note that due to the very small sample size (N = 3) and high variability between samples, t-test for significance only showed $p < 0.05$ for *Cd133*, *Hes1* and *Tgfα* (compared to control samples); however, all the markers showed upregulation with respect to control for all the samples. Because some of these factors can be expressed in a number of cells, including stellate cells and hepatocytes, their expression alone is not indicative of oval cell activation, but together with the oval marker expression data, it is highly suggestive of oval cell activation. In addition, upregulation of YAP/Notch related genes suggests the possibility of progenitor cell activation through these signaling pathways.

## Long-term regeneration of liver lobes following PEF ablation

Six months after the ablation, the IRE-ablated area of the liver appears macroscopically contracted (Fig. 7A). The histology showed that the ablated area regenerated, with a thin

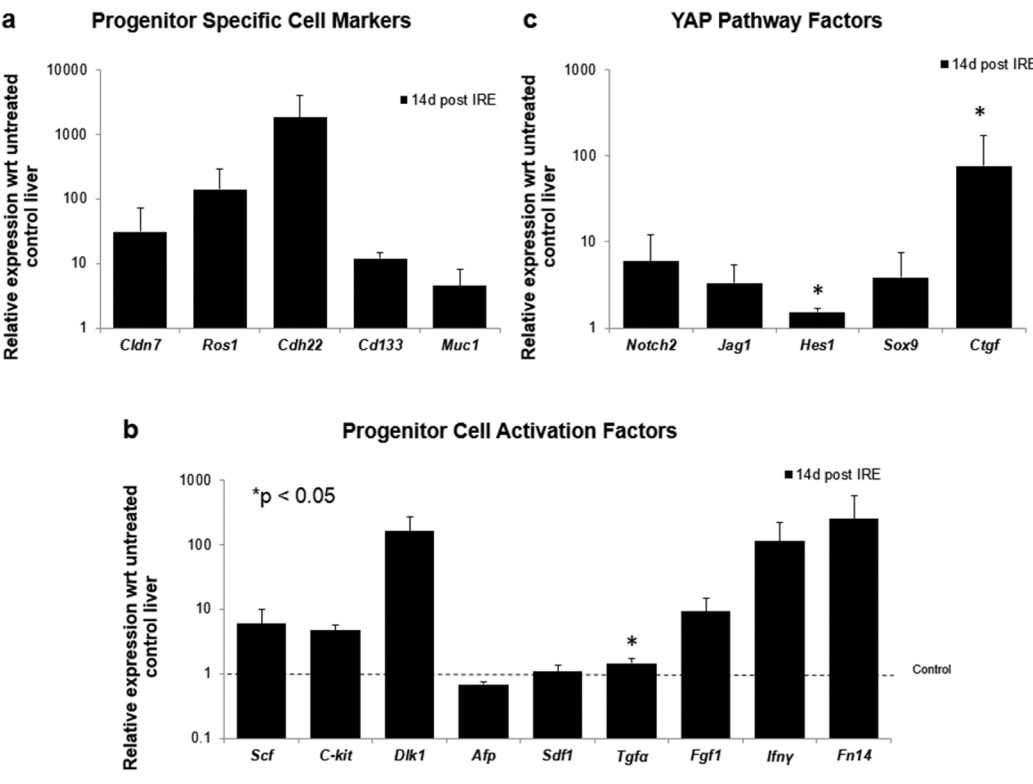

**Figure 6 Quantitative qRT-PCT of ablated tissue 2 weeks post-IRE.** 2 weeks after ablation, tissue biopsies of injured site were collected and qRT-PCR to analyze expression of oval cell markers and activation of pathways involved in oval cell regeneration. All oval cell specific markers were found to be upregulated relative to untreated tissue (A). In addition, a large number of genes related to oval cell activation were also found to be upregulated (B). Upregulation of Notch2 and downstream genes was observed in IRE treated samples compared to untreated controls (C). N = 3 for treatment and N = 2 for controls.

formation of fibrotic tissue (2–3 cells thick) in the treated area. Hepatocyte-like cells were present in all areas and liver structural components (vessels and ducts) were present (Fig. 7B). Importantly, no tumors were observed 6 months after the ablation, suggesting that IRE ablation did not lead to uncontrolled cell division.

## DISCUSSION

Irreversible electroporation (IRE) is an emerging minimally-invasive procedure for ablation of solid tumors, where tissues are ablated by high voltage pulsed electric fields. Current clinical trials involve primarily liver and pancreatic tumors (*Philips, Hays & Martin, 2013*). An interesting clinical finding suggests that IRE ablated tumors respond with significantly reduced fibrosis in comparison with standard radiofrequency ablation (*Narayanan, 2011*). We have previously shown that IRE ablation leads to scarless skin regeneration (*Golberg et al., 2013*). A unique effect of IRE is that it destroys the cell membranes, but preserves the extracellular milieu including extracellular matrix and microvascular structure (*Golberg et al., 2015a*) and in effect results in in vivo decellularization of the ablated area (*Phillips et al., 2012*). In this in vivo study in rats we found that IRE ablated zone in the liver regenerates without any scar formation as a result

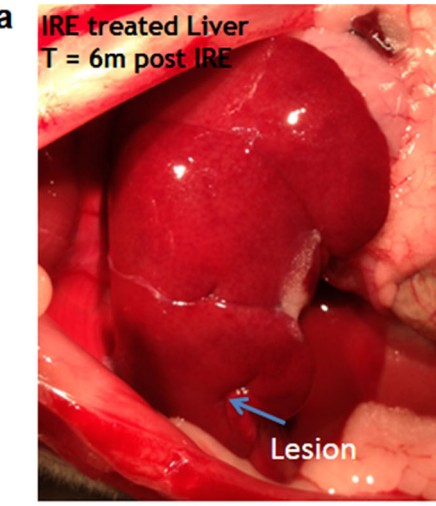

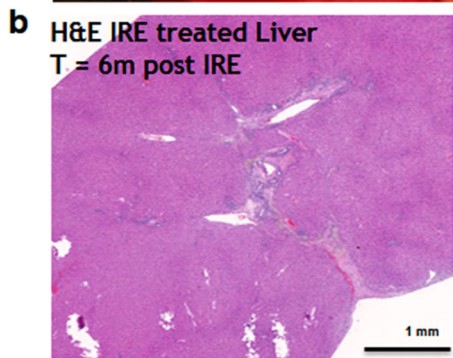

**Figure 7** **Liver regeneration 6 months after IRE ablation.** Small contraction is observed at the ablated site (A). The contraction is characterized by the intrusion of two-cell thick layer of connective tissue inside the regenerated part of the liver (B). No tumors or fibrotic tissue is observed at the regenerated zone. N = 6.

of both progenitor cell proliferation and differentiation at the center of ablated zone and by hepatocyte proliferation at the edges of the ablated zone, which are clearly demarcated.

The presence of a progenitor cell population within the liver has long been a topic of debate, but it is widely believed that these cells, termed oval cells, exist and repair injured livers when hepatocyte proliferation is impaired. Studies by *Yovchev et al. (2007)* identified six markers in rat livers that were unique to adult progenitor cells, all of which were found to be upregulated in IRE-treated tissues 2 weeks post-ablation. To further confirm activation of oval cells we looked at the expression of factors implicated in oval cell activation pathways, most of which were also shown to be upregulated. *Scf, Fgf1, Tgfα, Dlk1, Fn14* expressions are upregulated in liver injury induced by 2-acetaminofluorene and partial hepatectomy (AAF/PH), but not in that induced by partial hepatectomy alone, suggesting a role in progenitor dependent liver regeneration. Scf binds to C-kit whose expression has been shown to be restricted to oval cells (*Fujio et al., 1994*). Fgf1 expression is observed in oval cells, but also in basophilic hepatocytes, and hepatic stellate cells (*Lowes et al., 2003*). Similarly, Tgfα expression has also been observed in stellate cells after AAF/PH (*Evarts et al., 1993*). Dlk1 expression is observed in a subpopulation of

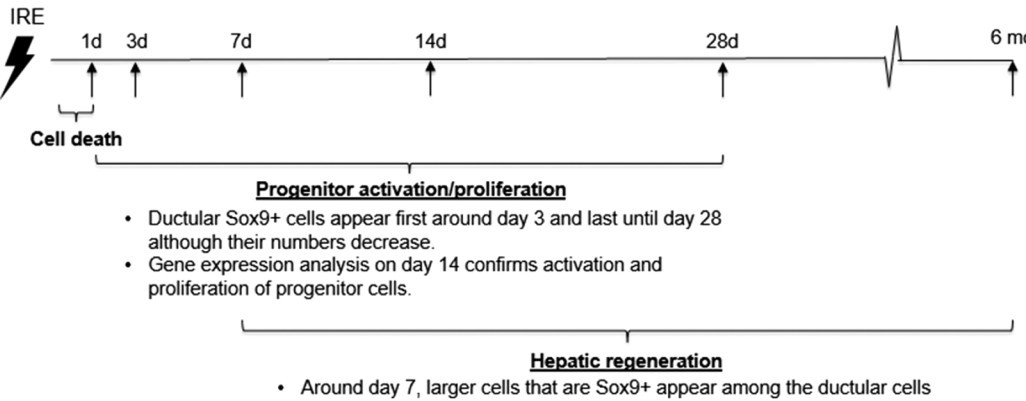

**Figure 8** Chronicle of events following rat liver ablation by irreversible electroporation.

oval cells found to reside distal to the periportal region and present morphology of small hepatocytes (*Tanimizu et al., 2004*). Fn14 expression has been observed in a few periductal cells of the adult liver where hepatic cells are suggested to reside, and binds to TNF-related weak inducer of apoptosis (TWEAK). It has been suggested that this interaction promotes oval cell proliferation, while no mitogenic effect from TWEAK is observed in hepatocytes (lacking Fn14 expression) (*Jakubowski et al., 2005*). Finally, while expression of Ifnγ is restricted to immune cells, it activates oval cell proliferation through activation of Jak/Stat pathway (*Akhurst et al., 2005*). Increased expression of these markers in our IRE treated samples (compared to control) suggests regeneration by oval cell activation.

An emerging theory suggests that a hepatocyte proliferation independent mechanism of liver regeneration is through dedifferentiation of hepatocytes to a progenitor stage. This occurs through inactivation of the Hippo pathway. Because Hippo pathway activity inhibits Yap, inactivation of Hippo results in Yap activation, and this was found to dedifferentiate adult hepatocytes, which differentiate again into hepatocytes upon reintroduction of Hippo. Because regulation of Yap is at the post-translational level, we studied activation of Yap through mRNA expression of *Ctgf*, as it is its most highly characterized target gene. In addition, it has been demonstrated that Yap activity leads to Notch pathway upregulation and this activity mediates hepatocyte dedifferentiation (*Yimlamai et al., 2014*). For this reason, we studied expression of *Notch* and its target genes and found that these genes are upregulated in IRE treated livers compared to normal livers. The chronicle of events that follow rat liver ablation by IRE is summarized in Fig. 8.

Chronic liver disease causes approximately 30,000 deaths per year in the US alone (*Habka et al., 2015*). The only definitive treatment for end stage liver disease is orthotopic liver transplantation and this is limited by the severe shortage of donor organs (*Habka et al., 2015*). In chronic liver disease, the liver cannot regenerate as usually seen in healthy livers because the hepatocyte proliferation is impaired and the hepatic progenitor cell proliferation and differentiation is not sufficient to achieve full recovery (*Best et al., 2013*). Previous studies achieved liver decellularization in vivo with IRE

and enhanced perfusion (*Sano et al., 2010*). Here we show that ablation of liver in vivo with IRE leads to 1) the activation and differentiation of the progenitor cells into hepatocyte like cells and 2) division of normal hepatocytes without formation of a tumor in the long term. These studies suggest a new approach to create a microenvironment inducive for progenitor-based liver regeneration and possibly functional liver recovery which would be a major advance in liver regenerative medicine.

Our results have additional implications in the clinical use of IRE in tumor ablation. In a recent review of a multi-institutional prospectively-collected registry of 150 patients undergoing 169 IRE ablations from 2009 through 2012, 31% of the patients had recurrence in the median follow up of 18 months. Of the total 31%, 10.7% were local recurrences at the ablated site (*Philips, Hays & Martin, 2013*). In previous work we have shown these recurrences could be driven by electric field sinks, resulting from the heterogeneous structure and conductivity of the liver (*Golberg et al., 2015b*). The results from this work indicate that the IRE treated extracellular matrix provides an environment for the activation and differentiation of progenitor cells. The role the tumor derived extracellular matrix on the cell differentiation is not completely understood, while some studies suggested the abnormal ECM promotes the formation of tumor microenvironment, tumor angiogenesis, lymphangiogenesis, inflammation and mostly interesting in the frame of this study-directly promoting cellular transformation and metastasis (*Lu, Weaver & Werb, 2012*). The role of the IRE spared tumor matrix in the follow up recurrences is an open question and requires further research. However, it is important to point out that our findings are valid only for the tested experimental conditions, which can be different in the clinical environment, where partially thermal effects of IRE have been observed (*Jarm et al., 2010*; *Kos et al., 2015*). In the case of a thermal effects, the regenerative pathways activated could be different.

This work is the first systematic long-term study of the liver response for the IRE ablation. Future work will focus on the lineage tracing of cells that contribute to regeneration. The source of cells that migrate to the IRE ablated area and then differentiate to hepatocyte like cells is currently unknown. Future work in this direction will focus on labeling liver cells, bone marrow cells and blood cells. Additional approaches could include transplantation of IRE decellularized liver from the female to the male rat with consequent chromosome staining. Additional studies will focus on the identification of the function of the finally differentiated hepatocyte like cells. Although by morphology the newly formed cells at the IRE ablated area resemble mature hepatocytes, their function is still to be determined.

## CONCLUSIONS

We have shown that the ablation of a normal liver by irreversible electroporation preserves the extracellular matrix, liver architecture and triggers the regenerative processes. Both progenitor cells and hepatocyte mitotic pathways were found to be active in the regeneration. Progenitor cells contribute to the liver regeneration at the center of the ablated zone. Hepatocyte mitosis is more dominant at the edges between the normal areas of the liver and the IRE ablated zone. Six months after the ablation no significant scars or

tumors have been observed, suggesting complete regeneration of the ablated area. The function of the newly formed hepatocyte-like cells is currently unknown and should be investigated in the future studies.

### Funding

This work was supported by the Claflin Distinguished Scholar Award from Massachusetts General Hospital (2013A050726) to Basak E. Uygun. The funders had no role in study design, data collection and analysis, decision to publish, or preparation of the manuscript.

### Grant Disclosures

The following grant information was disclosed by the authors:
Claflin Distinguished Scholar Award from Massachusetts General Hospital: 2013A050726.

### Competing Interests

B.E.U. has a financial interest in Organ Solutions, LLC, that is reviewed and arranged by Massachusetts General Hospital and Partners HealthCare in accordance with their conflict of interest policies. The rest of the authors of this manuscript have no conflicts of interest to disclose.

### Author Contributions

- Alexander Golberg conceived and designed the experiments, performed the experiments, analyzed the data, contributed reagents/materials/analysis tools, wrote the paper, prepared figures and/or tables, reviewed drafts of the paper.
- Bote G. Bruinsma conceived and designed the experiments, performed the experiments, analyzed the data, contributed reagents/materials/analysis tools, reviewed drafts of the paper.
- Maria Jaramillo conceived and designed the experiments, performed the experiments, analyzed the data, contributed reagents/materials/analysis tools, prepared figures and/or tables, reviewed drafts of the paper.
- Martin L. Yarmush conceived and designed the experiments, analyzed the data, contributed reagents/materials/analysis tools, reviewed drafts of the paper.
- Basak E. Uygun conceived and designed the experiments, analyzed the data, contributed reagents/materials/analysis tools, prepared figures and/or tables, reviewed drafts of the paper.

### Animal Ethics

The following information was supplied relating to ethical approvals (i.e., approving body and any reference numbers):

All animal procedures were approved by the Institutional Animal Care and Use Committee (IACUC) of the Massachusetts General Hospital and were kept in accordance with the guidelines of the National Research Council: Animal protocol #2013N000127.
## Data Deposition

The data generated in this research is supplied as a Supplemental Dataset file.

## Supplemental Information

Supplemental information for this article can be found online at http://dx.doi.org/
10.7717/peerj.1571#supplemental-information.

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
