# Peer review of "Rat liver regeneration following ablation with irreversible electroporation"

_PeerJ, doi:10.7717/peerj.1571_

## Round 0.1 · original submission · Minor Revisions

The manuscript has some interesting and new information on IRE in liver regeneration. This is applicable for further mechanical approaches.

Additionally, such regenerated liver tissues can be explored for the cell-cell interaction signaling.

Please address the reviewer comments in your revision and rebuttal letter.

Reviewer 1 ·

Basic reporting

The paper by Goldberg et al presents a compelling study which demonstrates that IRE provides the suitable microenvironment for tissue regeneration. The researchers show that IRE induces rapid decellularization while leaving the underlying architecture intact and demonstrate liver regeneration. The proposed model has tremendous implications in the field of tissue engineering as well as to study tissue engineering.

Experimental design

The experimental methods are technically sound.

A few discretionary comments:
- If anything, it would be nice to have a little more detail regarding the thermal effects rather than need to just look at another paper (e.g., ...chosen to diminish thermal effects (<XXC temperature increase) and was based on our previous optimization studies (Goldberg...)
- I think the concept of the "electric field sink" is important for clinicians and could be explained further.
- I'm not sure if many of the readers will know what Tweak is.

Validity of the findings

The results the authors conclude from their findings are substantiated. Previous studies by their group and others have shown that IRE can cause decellularization. To the best of my knowledge, this is the first to demonstrate activation/differentiation of progenitor cells into hepatocyte cells and division of normal hepatocytes.

Additional comments

I believe the manuscript would be of great interest to the electroporation community, tissue engineering and those who study liver disease.

·

Basic reporting

The authors present a manuscript, where they follow rat liver regeneration after complete ablation with irreversible electroporation. Irreversible electroporation is an emerging technique for soft tissue ablations, which is used in the treatment of liver, pancreatic, kidney, prostate and lung cancers, often using minimally invasive procedures under radiological guidance. As a physical method, which is however not based on a thermal effect, IRE is very promising even in close proximity of blood vessles which limit thermal methods. Although the use of IRE is on the rise, the basic mechanisms underlying the action of the method, its cell killing pathways and post-treatment regeneration are still not entirely elucidated. Therefore, the present manuscript presents important data for the study of IRE and brings solid support for the oft claimed, but as of yet not fully proven sparing of extracellular matrix.
The present manuscript presents a very detailed tracking of the tissue regeneration following IRE ablation at six time-points from one day to 6 months after the procedure. This presents a as of yet unpublished temporal resolution, and the authors also back that up with extensive staining, immunohistochemistry and q-RT-PCR. This is good evidence for the mechanisms in action.
While I would recommend the manuscript for publication, I have some minor issues, that I would like to see addressed, which however should not require additional experiments.

Experimental design

The pulse pulsing protocol, electrodes used and the description of the choice of pulses should in my opinion belong in the methods section. I would also like to see more detailed description of the electrodes. Although it might be present in previous publications, I still believe it should be at least repeated here. Pertaining to that, I think the results should include measurements of the size of the ablation zones. Although the depth should be 3 mm according to the electrode separation, at least the dimension in the plane of the histological slices, but perpendicular to the electric field should also be given. Also, the pulsing protocol seems to be relatively aggressive, considering that the electrode setup has a more plate-like setup, which means that the electric field between the electrodes should be very homogeneous. According to previously published data, ablation should be achieved at roughly half the voltage that the authors used. This should mean that the ablation zone should also extend quite a bit outside from the cylindrical volume between the electrodes. Unfortunately, it is currently not possible to judge how much.

Validity of the findings

I am not entirely qualified to review histological and immunohistochemical data, so another reviewer should comment on these.
In the first paragraph of the results, the authors note that tissue lost normal color and was distinctly white in appearance. This would be expected from the immediate abrogation of blood flow that electroporation causes in tissue, and is a known effect. It occurs immediately after pulse delivery and persists for hours afterwards already at much lower field strengths (Jarm et al. 2010).

My last comment is, that the pulsing protocols used in clinical practice in humans are often very much thermal, as can be seen for example in (Kos et al. 2015). This means, that although the data presented here is important, valid and useful, it cannot directly be applied to human IRE trials, since the pulse protocols there do cause significant heating and therefore could be activating completely different pathways to the ones presented here. This should be clearly stated in the discussion that these findings only apply where decisively nonthermal protocols are used.

Jarm, Tomaz, Maja Cemazar, Damijan Miklavcic, and Gregor Sersa. 2010. “Antivascular Effects of Electrochemotherapy: Implications in Treatment of Bleeding Metastases.” Expert Review of Anticancer Therapy 10 (5): 729–46. doi:10.1586/era.10.43.
Kos, Bor, Peter Voigt, Damijan Miklavcic, and Michael Moche. 2015. “Careful Treatment Planning Enables Safe Ablation of Liver Tumors Adjacent to Major Blood Vessels by Percutaneous Irreversible Electroporation (IRE).” Radiology and Oncology 49 (3): 234–41. doi:10.1515/raon-2015-0031.

Additional comments

I’ll leave the next suggestions completely to the authors, but I feel that the manuscript could be nicely improved by adding an illustrated timeline of the experiment with the most significant events illustrated along a description of the changes during each time point. I feel, that this could really make the paper shine, although it is not strictly necessary in terms of data completeness.

Finally, the fist panel of figure 1 is an eyesore in an otherwise very nicely illustrated manuscript. It would be really nice if it was redrawn to be slightly more informative, better looking and generally more professional. What does that black rounded rectangle connected to the wires represent? I have an idea, but it is not necessarily the correct one.

Reviewer 3 ·

Basic reporting

The manuscript presents a study on the use of IRE to ablate a section of rat liver tissue in vivo and the subsequent repopulation of the ablated area by progenitor cells and hepatocytes. The study presents promising results on the potential participation of progenitor cells to migrate to the ablated area and differentiate into hepatocytes to regenerate the region of dead tissue. The results, however, lack statistical significance due to the small sample size available and high variability obtained. In spite of this, the manuscript is recommended for publication after the comments below have been addressed. It is understandable that not all scientific studies will have an ideal sample size to achieve statistical significance but the authors should be extra careful about how the data is presented and conclusions are drawn as to not make unfounded claims that can mislead readers.
- The title should be reworded as it can be misleading in its current form. Rat livers were not completely ablated with IRE and subsequently regenerated; only a small portion was ablated.
- Is the change in permeability really irreversible? The study referenced for this claim (Golberg and Yarmush 2013) does not seem to show results to prove this.
- Line 46, what is meant by “similar cell-ablative effects”? Is it the mechanism of cell death, volume of ablated tissue, tissue response? Please be more specific here.
- Line 47, if the mechanism of cell death remains unclear wouldn’t this disprove the previous claim of the change in permeability being irreversible? Please revise and make pertinent changes.

Experimental design

- Include p-values for figure 6. It is important to point out the results that are statistically significant and those that are not as to not mislead the reader. This is mentioned in the text but it should also be clearly represented in the graphs shown.

Validity of the findings

- How can the similarity in Sox9 expression at days 3 and 28 be explained? (fig 5)
- Line 233, the statement “IRE ablated liver regenerates completely” can be misleading since the study only evaluated a small zone of ablated tissue and the subsequent cell migration and proliferation to the area by both hepatocytes and progenitor cells. Further characterization of the cells in the area, including their function, need to be performed before such a claim can be made. Please reword.
- Lines 311-312: once again here it is stated that “a complete ablation of normal liver by IRE” is achieved which does not represent what was actually done in the study and can be misleading. The liver was not completely ablated, only a small area or a couple of small areas on each liver. Please reword to present an accurate recount of the study.

Additional comments

- Line 73, “is occurs” remove “is”
- Check for typos in figure 3 caption
- Lines 202 to 204, check and correct for typos. What is IHC? Immunohistochemistry is the guess but this is not mentioned anywhere in the manuscript. Please include earlier in the manuscript or supply a list of acronyms with submission.
- Markers Afp and Ctgf are not listed in parentheses from line 198 to 201, but are later discussed within text. Please include in the respective lists.
- Figure 5 caption, move “(b)” to come right after “28 days after ablation”. Current location can be confusing to the reader.
- What is AAF/PH model and PH model? Lines 243, 244. Include the full name (2-acetaminofluorene/partial hepatectomy) somewhere in the manuscript or as part of a list of acronyms.
- Line 286, “this these” remove “this”

---

## Round 0.2 · accepted · Accept

No more comments. The work is valuable for further justification in field of tissue regeneration.